# Mechanism of phosphate sensing and signaling revealed by rice SPX1-PHR2 complex structure

Jia Zhou [1,2,7], Qinli Hu [1,2,7], Xinlong Xiao[1], Deqiang Yao[3], Shenghong Ge[1,2], Jin Ye [4], Haojie Li[1], Rujie Cai[5], Renyang Liu[1], Fangang Meng[6], Chao Wang [4], Jian-Kang Zhu[1], Mingguang Lei [1✉] & Weiman Xing [5✉]

Phosphate, a key plant nutrient, is perceived through inositol polyphosphates (InsPs) by SPX domain-containing proteins. SPX1 an inhibit the PHR2 transcription factor to maintain Pi homeostasis. How SPX1 recognizes an InsP molecule and represses transcription activation by PHR2 remains unclear. Here we show that, upon binding InsP$_6$, SPX1 can disrupt PHR2 dimers and form a 1:1 SPX1-PHR2 complex. The complex structure reveals that SPX1 helix α1 can impose a steric hindrance when interacting with the PHR2 dimer. By stabilizing helix α1, InsP$_6$ allosterically decouples the PHR2 dimer and stabilizes the SPX1-PHR2 interaction. In doing so, InsP$_6$ further allows SPX1 to engage with the PHR2 MYB domain and sterically block its interaction with DNA. Taken together, our results suggest that, upon sensing the surrogate signals of phosphate, SPX1 inhibits PHR2 via a dual mechanism that attenuates dimerization and DNA binding activities of PHR2.

[1] Shanghai Center for Plant Stress Biology and Center of Excellence in Molecular Plant Sciences, Chinese Academy of Sciences, Shanghai 200032, China. [2] University of Chinese Academy of Sciences, Beijing 100049, China. [3] State Key Laboratory of Oncogenes and Related Genes, Ren Ji Hospital, Shanghai Jiao Tong University School of Medicine, Shanghai 200127, China. [4] MOE Key Laboratory for Membrane-less Organelles & Cellular Dynamics, Hefei National Laboratory for Physical Sciences at the Microscale, School of Life Sciences, Division of Life Sciences and Medicine, University of Science and Technology of China, Hefei 230027, China. [5] Shanghai Key Laboratory of Plant Molecular Sciences, College of Life Sciences, Shanghai Normal University, Shanghai 200234, China. [6] Beijing Neurosurgical Institute, Beijing Tiantan Hospital, Capital Medical University, Beijing 100070, China. [7] These authors contributed equally: Jia Zhou, Qinli Hu. ✉email: mglei@cemps.ac.cn; weimanxing@shnu.edu.cn

Phosphorus is a fundamental element of all living organisms and represents a key building block of many cellular molecules, such as the genetic material DNA and the energy carrier ATP[1]. Therefore, all cells need to maintain a sufficient concentration of phosphate (Pi) in their cytoplasm[2]. For plants, the availability of inorganic Pi in soil is poor due to its low mobility[3,4]. To cope with the fluctuating levels of Pi, plants have evolved sophisticated strategies to modulate Pi uptake and remobilization[5–9]. The PHR transcription factors are the central regulators of Pi signaling, which bind to the PHR1-binding sequence (P1BS) in the promoter regions of phosphate starvation-induced (PSI) genes[10]. By activating the expression of PSI genes, PHRs enhance Pi uptake under Pi-deficient conditions[11,12]. To avoid Pi toxicity caused by excessive Pi accumulation in plants, SPX proteins, on the other hand, bind to PHR proteins and inactivate PHR-induced transcription under Pi-replete conditions[13,14].

How plants sense cellular Pi levels to manipulate PSI genes expression induced by PHR2 is a fascinating and key question in Pi homeostasis. The PHR proteins belong to the MYB–coiled-coil (MYB-CC) family, which is characterized by a conserved MYB DNA-binding domain and a potential CC dimerization domain[11,15,16]. In fact, dimerization of *Arabidopsis* PHR1 has been shown to be important for its high-affinity DNA binding function[12]. Previous studies indicated that rice SPX1 interacts with PHR2 in a Pi-dependent manner and acts as an inhibitor to repress PHR2 transcription activation under Pi-replete conditions[17]. These findings implied the important roles of SPX domain-containing proteins in sensing Pi. Recent structural analysis of the SPX-domain, in conjunction with in vivo studies, has provided compelling evidence supporting the functions of the SPX domain as Pi sensors. Interestingly, SPX domains do not appear to sense inorganic Pi directly, but instead recognize soluble inositol polyphosphates (InsPs), such as $InsP_7$ or $InsP_8$, which act as a proxy for the cellular Pi status[18,19]. Such a mechanism has recently been unveiled by the crystal structures of SPX domains in complex with $InsP_6$, a commercially available substitute of $InsP_7$ and $InsP_8$[18,20–22]. Despite these recent advances, how SPX domain proteins recognize the InsP molecules and regulate downstream players in Pi homeostasis remains unclear. Here, we analyzed the effects of $InsP_6$ binding to rice SPX1-PHR2 complex and determined the $InsP_6$-bound SPX1-PHR2 complex structure. Together with biochemical studies, our structure unravels a unique dual mechanism by which SPX1 mediates Pi sensing and signaling.

## Results

**$InsP_6$-induced SPX1-PHR2 heterodimer formation**. Due to the instability of bacteria-produced full length PHR2 and SPX1 proteins, we purified an MYB and CC domain-containing PHR2 fragment ($PHR2^{230–426}$), which has been previously reported to interact with full length and a C-terminally truncated fragment of SPX1 ($SPX1^{1–259}$) (Supplementary Fig. 1a)[17]. To verify whether InsP molecules mediate SPX1-PHR2 complex formation, we used gel filtration assay to characterize the $SPX1^{1–259}$-$PHR2^{230–426}$ interaction. In contrast to no peak shift of PHR2 in absence of $InsP_6$, the peak of $PHR2^{230–426}$ proteins on the column shifted later and co-migrated with $SPX1^{1–259}$ proteins in presence of $InsP_6$, indicative of $InsP_6$-induced specific interaction (Fig. 1a). Interestingly, the elution volume of PHR2 shifted from 12.2 to 13.2 ml when forming a complex with SPX1, indicative of changes of the PHR2 dimer state. To verify whether PHR2 forms a dimer in solution as reported in the literature[12], size exclusion chromatography with inline multi-angle light scattering (SEC-MALS) and crosslinking experiments were applied to measure the molecular weight of $PHR2^{230–426}$ and evaluate its oligomeric

state. Isolated $PHR2^{230–426}$ appeared to exist in solution as a mixture of dimer and tetramer species based on both experiments (Fig. 1a and Supplementary Fig. 2a). Similar analysis of *Arabidopsis* $PHR1^{227–358}$, which contains the MYB and CC domains only, yielded a molecular weight of 29.2 kDa, indicative of a dimeric form (Supplementary Fig. 2b). We speculated that dimeric PHR2 was probably more prevalent in vivo and its tetrameric form might be detected in vitro due to nonspecific interactions. Based on the facts that SPX1-PHR2 complex was eluted later than PHR2 itself on the gel filtration column and that PHR2 might be a dimer in solution, whereas $SPX1^{1–259}$ is monomeric, we conclude that the $SPX1^{1–259}$-$PHR2^{230–426}$ complex most likely contains one copy of $SPX1^{1–259}$ and $PHR2^{230–426}$ each. Indeed, SEC-MALS analysis indicated that the $InsP_6$-induced SPX1-PHR2 complex was eluted with a molecular mass of 50.1 kDa (Fig. 1a). Hence, SPX1 and PHR2 formed a complex with a 1:1 stoichiometry in the presence of $InsP_6$ and the PHR2 dimer was disrupted during the complex formation. It is noticed that SPX1 was slightly shifted by PHR2 as detected by size exclusion chromatography in the absence of $InsP_6$ (Fig. 1a). However, it appeared to dynamically interact with the transcription factor but failed to form a stable and homogenous 1:1 heterodimer with PHR2.

If the stable binding of SPX1 to PHR2 hinges on PHR2 monomerization, a dimerization-defective mutant of PHR2 should be able to form a complex with SPX1 even in the absence of $InsP_6$. Based on a recently reported *Arabidopsis* PHR1 CC dimer structure[20], we introduced several mutations that were expected to disrupt the PHR2 dimer interface. All PHR2 mutants, L348A, L358A, I362A, and L344R/Q351R/Q355R abolished the dimer formation in gel filtration and SEC-MALS assays (Fig. 1b). Strikingly, these PHR2 monomeric mutants, represented by the triple mutant (L348A/L358A/I362A, $PHR2^{3M}$) could bind SPX1 with a Kd of ~0.17 μM in an $InsP_6$-independent manner by gel filtration, SEC-MALS, and ITC assays (Fig. 1c, d). The binary complex was eluted from a gel filtration column at a similar position as the SPX1-PHR2 complex assembled by $InsP_6$ and shared almost the same molecular weight of about 47.0 kDa.

Using ITC assay, we further analyzed the effect of $InsP_6$ on the DNA binding activity of PHR2. Individual $InsP_6$ or SPX1 did not affect the ability of PHR2 to bind DNA. However, when both $InsP_6$ and SPX1 were mixed with PHR2, the DNA binding activity of the transcription factor was completely abolished (Fig. 1e). In line with the documented importance of PHR2 dimer[12], we noticed that PHR2 monomeric mutant exhibited lower DNA binding and transcriptional activation capability compared with wild type in ITC and dual-LUC transient transcriptional activity assays (Fig. 1e, f). In contrast to the wild type PHR2, the transcriptional activity of the PHR2 monomeric mutant could be inhibited by SPX1 under Pi starvation conditions in accordance with its constitutive interaction with SPX1. Based on these results and our aforementioned analysis shown in Fig. 1a, we conclude that PHR2 monomerization is a critical step in SPX1- and $InsP_6$-induced inactivation of the transcription factor.

**Overall structure of SPX1-PHR2**. The SPX1-PHR2 complex formation mediated by InsPs initiates downstream transcriptional inhibition of PSI genes. To understand how SPX1 responds to InsP binding, we determined the $InsP_6$-SPX1-PHR2 complex structure at 2.6 Å resolution using single-wavelength anomalous diffraction. The model was refined to a final $R_{work}$ and $R_{free}$ value of 22.6 and 27.4%, respectively (Supplementary Table 1). There are two amino acids located in the outlier region of Ramachandran plot probably due to poor accuracy associated with the electron density in the local region. The final model of the $InsP_6$-SPX1-PHR2 complex contains two copies of the $InsP_6$-SPX1-

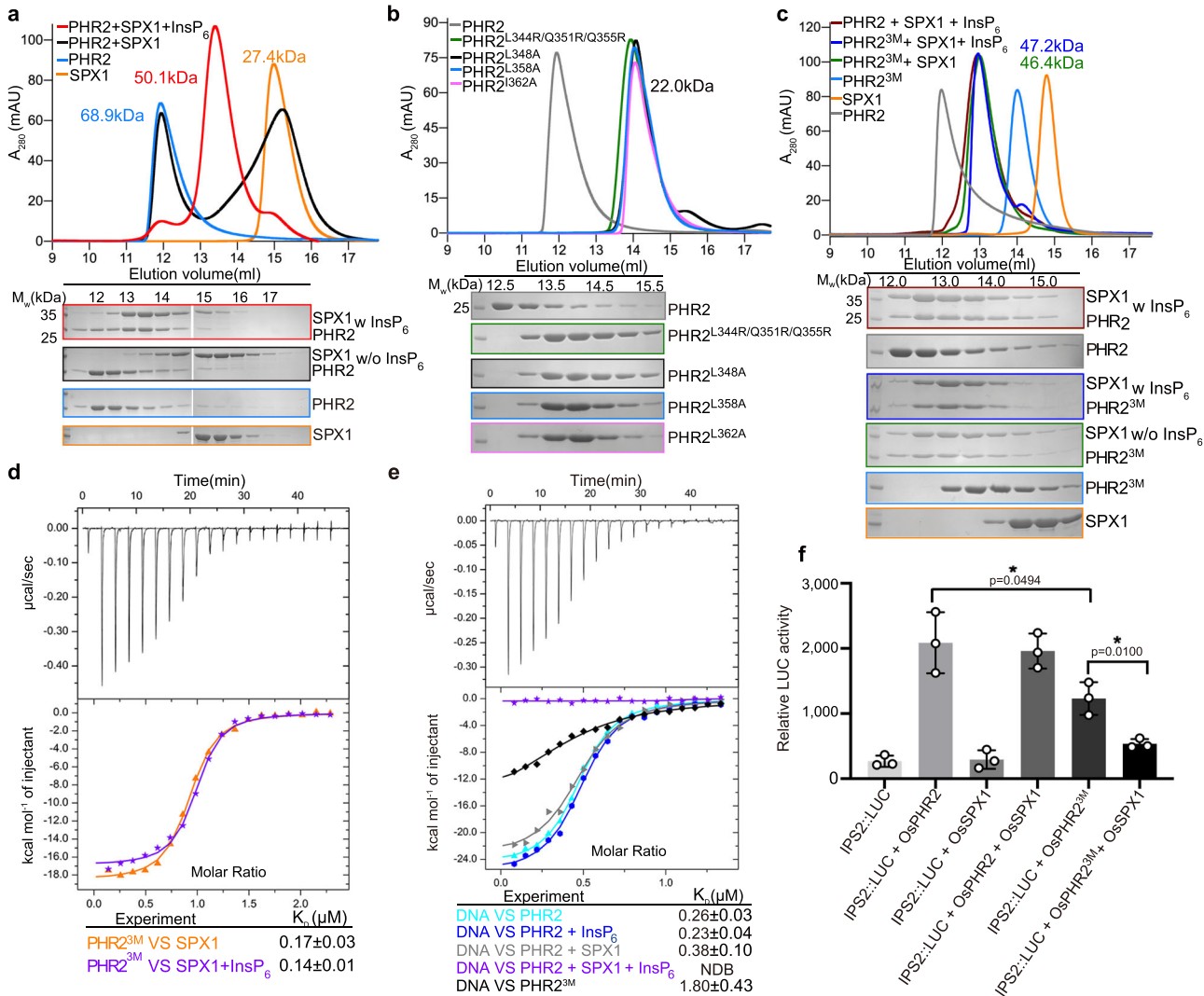

**Fig. 1 Monomeric SPX1-PHR2 complex triggered by InsP$_6$ was determinant for PHR2-DNA binding inhibition. a** SPX1$^{1-259}$-PHR2$^{230-426}$ complex formation was induced by InsP$_6$, as identified by gel filtration assay. (Upper) Gel filtration profiles of SPX1$^{1-259}$, PHR2$^{230-426}$, SPX1$^{1-259}$ and PHR2$^{230-426}$ in the presence or absence of InsP$_6$ were color-coded. (Lower) Coomassie-blue stained SDS-PAGE gels of peak fractions. The molecular weights were labeled based on the SEC-MALS results. Experiments were independently repeated three times with similar results. Uncropped gel images are available as source data. **b** Mutations of PHR2$^{230-426}$ disassembled the PHR2$^{230-426}$ dimer into monomer. The molecular weights were measured by SEC-MALS. Experiments were independently repeated three times with similar results. Uncropped gel images are available as source data. **c** Monomeric mutant of PHR2$^{230-426}$ L348A/L358A/I362A(PHR2$^{3M}$) could interact with SPX1$^{1-259}$ in absence of InsP$_6$. The molecular weights were measured by SEC-MALS. Experiments were independently repeated three times with similar results. Uncropped gel images are available as source data. **d** Binding affinity of PHR2$^{3M}$ with SPX1$^{1-259}$ in the presence (purple) or absence of InsP$_6$ (orange) were detected by ITC. **e** Effects of SPX1, InsP$_6$, and PHR2$^{3M}$ on PHR2-DNA(P1BS) binding affinity as detected by ITC. ITC curves for the titration of DNA (P1BS) with PHR2$^{230-426}$ (cyan); PHR2$^{230-426}$ and InsP$_6$ (blue); PHR2$^{230-426}$ and SPX1$^{1-259}$ (gray); PHR2$^{230-426}$, SPX1$^{1-259}$ and InsP$_6$ (purple); PHR2$^{3M}$ (black). **f** Dimerization was important for the transcription activity of PHR2, and SPX1 can repress transcription activity of PHR2$^{3M}$ in *phr1 phr2* mutant protoplast under Pi-deplete condition. Values are means ± s.d. of three independent biological replicates. Sets of data were compared by paired-samples T test (two-tailed), where the asterisk indicates statistically significant differences (*$P < 0.05$).

PHR2 complex in the asymmetric unit (Supplementary Fig. 1b). A highly crystallizable T4 lysozyme (T4L) was fused to the C-terminus of SPX1 to facilitate crystallization and improve the quality of crystals. The two T4 lysozyme molecules in the asymmetric unit adopt different orientations with respect to SPX1 due to crystal packing (Supplementary Fig. 1c).

The SPX domain of SPX1 consists of two long core helices α3 and α4, and two shorter C-terminal helices, α5 and α6. Two N-terminal helices α1 and α2 form a helical hairpin. The two SPX1 molecules in the asymmetric unit, which share high structural homology with documented SPX domain structures, can be superimposed with each

other with a Cα RMSD of 1.36 Å (Supplementary Fig. 1d, e)[18]. The PHR2 MYB domain contains three consecutive α helices, which give rise to the conserved fold of the MYB proteins. The MYB domains of rice PHR2 and *Arabidopsis* PHR1 can be superimposed with a RMSD of 1.40 Å (Supplementary Fig. 1f)[23]. The PHR2 CC motif long helix closely aligns with the CC motif of *Arabidopsis* PHR1 with a Cα RMSD of 0.89 Å (Supplementary Fig. 1g)[20]. Interestingly, both the MYB domain and CC motif of PHR2 make direct contact with SPX1. They approach SPX1 from two different directions and sandwich SPX1 in between with extensive protein–protein interfaces (Fig. 2a).

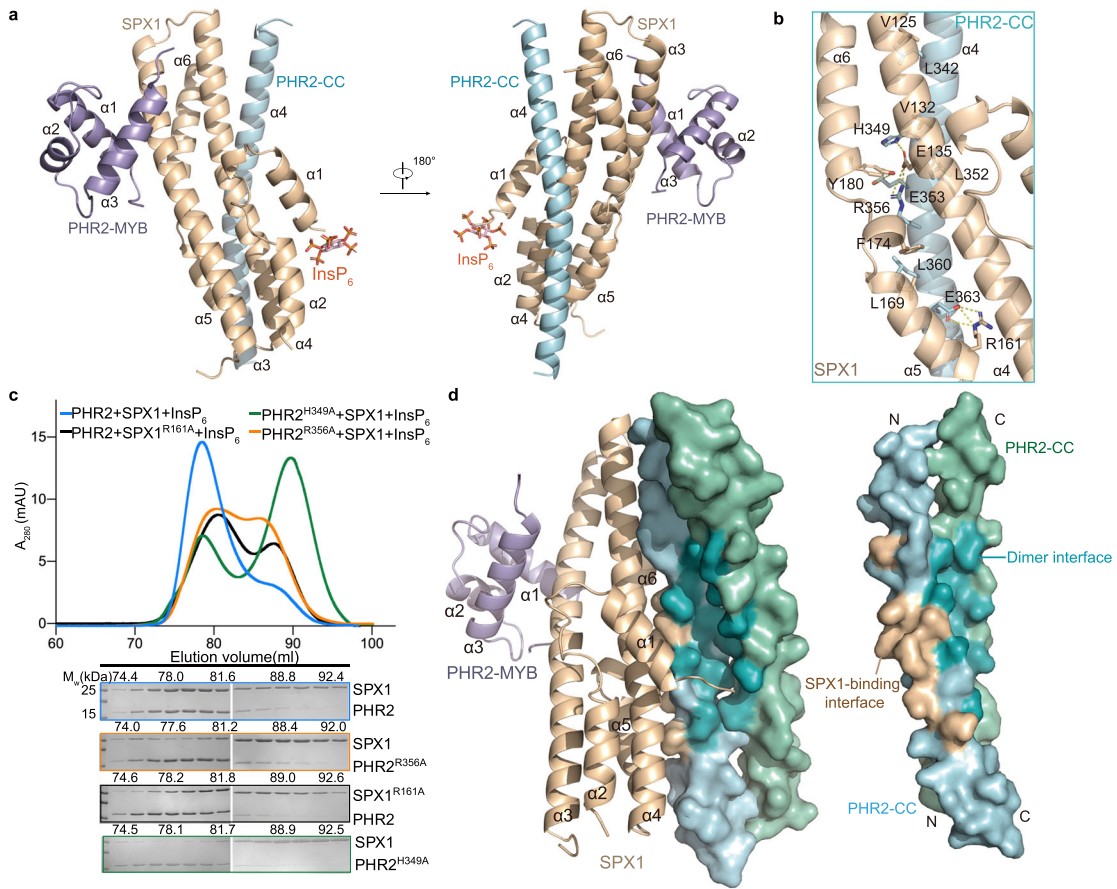

**Fig. 2 SPX1-PHR2 CC interface is compatible with PHR2 dimer interface. a** The crystal structure of $InsP_6$-SPX1-PHR2 complex. SPX1 was colored in wheat, and the MYB and CC domains of PHR2 were colored in light blue and pale cyan respectively. $InsP_6$ was represented as a stick model in red. $InsP_6$-SPX1-PHR2 complex was shown in two opposite views. The secondary structure elements and $InsP_6$ molecule were labeled. **b** Close-up view of SPX1-PHR2 CC domain interface. Critical interface residues were shown as sticks and labeled. The yellow dashed lines represent hydrogen bonds involved in the interaction of SPX1 and PHR2. **c** Mutations compromised the SPX1[1–198]-PHR2[248–380] complex formation. (Upper) Gel filtration profiles of SPX1[1–198] and PHR2[248–380] were color-coded to show different SPX1[1–198] mutants and PHR2[248–380] mutants. (Lower) Coomassie-blue stained SDS-PAGE gels of peak fractions. Experiments were independently repeated three times with similar results. Uncropped gel images are available as source data. **d** SPX1-PHR2 CC interface does not interfere with PHR2 dimerization. (Left) Superimposition $InsP_6$-SPX1-PHR2-ternary complex structure onto AtPHR1 CC (PDB:6TO5) dimer structure. (Right) Surface representation of rice PHR2 CC dimer model. SPX1-binding interface was colored in wheat and PHR2 dimer interface was colored in teal.

**SPX1-PHR2 CC motif interface**. SPX1 interacts with the PHR2 CC motif via an extensive and elongated interface. Specifically, the PHR2 CC motif long helix interacts with three helices of SPX1 (α4, α5, and α6) through several ionic interactions and hydrogen bonds (Fig. 2b). SPX1 E135 and R161 form salt-bridges with PHR2 R356 and E363 respectively. In addition, SPX1 E135, R161, and Y180 interact with PHR2 H349, E363, E353, and R356 through hydrogen bonds.

To verify the structural observations, we individually mutated residues in the SPX1-PHR2 CC motif interface into alanine. The R161A mutation in SPX1 and the PHR2 mutants, H349A and R356A, all compromised SPX1-PHR2 stable complex formation, whereas other SPX1 mutants, E135A and Y180A, and PHR2 mutants, E353A and E363A, failed to alter SPX1-PHR2 binding (Fig. 2c and Supplementary Fig. 3b, d). Moreover, hydrophobic residues, such as V125 and V132 in SPX1 and L352, L342 of PHR2 also contribute to the interface formation (Fig. 2b). Substitution of V125 and V132 in SPX1 and L342, L352 in PHR2 with alanine abolished SPX1-PHR2 interaction (Supplementary Fig. 3a, b). The L360 of PHR2 looks like participate the hydrophobic patch formation, however, PHR2 single mutation

L360A retained the ability to bind SPX1. Collectively, SPX1-PHR2 CC interface, which is largely stabilized by hydrogen bonds, salt-bridges and hydrophobic interactions, is critical for complex formation.

The crystal structure of the Arabidopsis PHR1 CC dimer has recently been determined. Surprisingly, superposition analysis indicates that the PHR CC dimer interface is fully compatible with SPX1 binding (Fig. 2d), suggesting that the SPX1 binding via the SPX1-PHR2 CC interface itself does not interfere with PHR2 dimerization. In order to disrupt the PHR2 dimer, $InsP_6$ must act through a mechanism outside the SPX1-PHR2 CC interface.

**$InsP_6$-binding site**. Our structure reveals three $InsP_6$ molecules in the asymmetric unit (Supplementary Fig. 4a). One $InsP_6$ molecule participates in crystal-packing interactions, while the other two are each captured by an SPX1 molecule via a highly basic surface groove formed between helices α1, α2, and α4 (Figs. 2a, 3a and Supplementary Fig. 4c). Importantly, this binding mode of these two $InsP_6$ molecules to SPX1 is very similar to the reported $InsP_6$-SPX domain structures

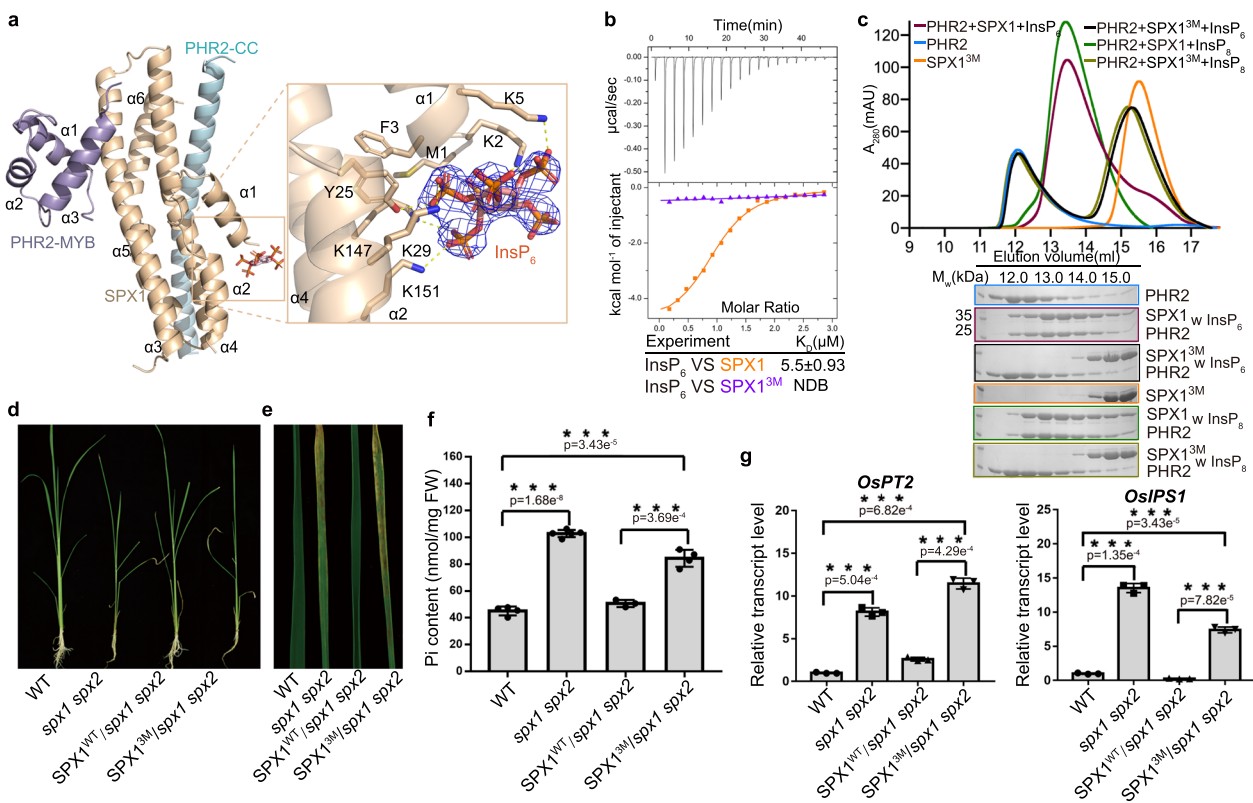

**Fig. 3 Mutations in SPX1 InsP₆-binding sites affect SPX1-PHR2 complex formation and Pi signaling. a** A close-up view of the SPX1-InsP₆ interactions. InsP₆ was represented as a stick model with a stimulated annealing omit map and contoured at 1.1σ. Residues of SPX1 interacting with InsP₆ were shown as sticks and labeled. The yellow dashed lines represent hydrogen bonds involved in the interaction of SPX1 and InsP₆. **b** Binding affinities of InsP₆ with SPX1$^{1-259}$ (orange) and SPX1$^{1-259\ Y25F/K29A/K151A}$ (SPX1$^{3M}$, purple) were measured by ITC. NDB, no detectable binding. **c** Mutations of InsP₆-binding sites in SPX1$^{1-259}$ impaired the SPX1$^{1-259}$-PHR2$^{230-426}$ interaction. The assays were performed as described in (Fig. 1a). Experiments were independently repeated three times with similar results. Uncropped gel images are available as source data. **d** Growth phenotype of WT, *spx1 spx2*, and two kinds of transgenic plants were grown under Pi-sufficient condition (SPX1$^{WT}$/*spx1 spx2* and SPX1$^{Y25F/K29A/K151A}$/*spx1 spx2*). **e** Phenotype of phosphate excess in leaves of plants shown in (**d**). **f** Pi contents of plants shoots shown in (**d**). Error bars, mean ± s.d. The number of biologically independent samples of WT, *spx1 spx2*, SPX1$^{WT}$/*spx1 spx2* and SPX1$^{3M}$/*spx1 spx2* is 4, 5, 3 and 4, respectively. Sets of data were analyzed by independent-samples T test (two-tailed), where *** represents a statistically significant difference at $p < 0.001$. **g** PSI genes (*OsIPS1* and *OsPT2*) expression in the root of the plants shown in (**d**). Error bars, mean ± s.d. The bar graphs show the results from three biologically independent samples ($n = 3$). Sets of data were compared by paired-samples T-test (two-tailed), where *** represents a statistically significant difference at $p < 0.001$.

(Supplementary Fig. 4b). Three lysine residues (K29, K147, and K151) and a tyrosine residue (Y25) from SPX1 contribute to the interaction with InsP₆ (Fig. 3a and Supplementary Fig. 5). Their functional importance is underscored by their conservation among SPX domains from different eukaryotes (Supplementary Fig. 6a). To validate the roles of these conserved amino acids in InsP₆ binding, a triple mutation of SPX1, Y25F/K29A/K151A, was designed and tested in an ITC assay. In contrast to the moderate InsP₆ binding ability of the wild-type protein, the SPX1 mutant lost its InsP₆-binding activity, confirming the importance of these three residues for InsP₆ perception (Fig. 3b). SPX1 K147A mutation had little effect on InsP₆ binding affinity (Supplementary Fig. 7).

Consistent with the notion that InsP₆ mediates the disruption of the PHR2 dimer outside the SPX1-PHR2 CC interface, our structure shows that the SPX1 InsP₆-binding site is removed from the long PHR2 CC helix. The closest residue of PHR2, Q359, is about 13 Å away from InsP₆, which nevertheless is required for the stable SPX1-PHR2 interaction as shown in our size exclusion chromatography analysis. To further confirm the critical role of InsP molecules in stabilizing the 1:1 SPX1-PHR2 heterodimeric complex, we measured the binding of the SPX1 Y25F/K29A/K151A triple mutant with PHR2 using gel filtration assay. As

shown in Fig. 3c, the SPX1 triple mutant could not co-migrate with PHR2 despite the presence of InsP₆ or InsP₈, indicative of weak or no binding between SPX1 triple mutation and PHR2. Importantly, rice SPX1 Y25F/K29A/K151A triple mutant could not rescue *spx1spx2* double mutant phenotype (Fig. 3d, e). Not only the Pi content in the triple mutant was increased compared with wild type, PSI genes were also up-regulated (Fig. 3f, g). These data support the idea that the binding of InsP₆ at this allosteric site somehow enables SPX1 to interact with and inhibit PHR2.

In the crystal, we notice that the SPX1 helix α1 residues M1, K2, F3, and K5 also participate in interacting with InsP₆ (Fig. 3a and Supplementary Fig. 5). Deletion of the N-terminal helix α1 (ΔN17) reduced InsP₆ binding affinity by ~6-fold (Fig. 4a). But this N-terminally truncated SPX1 mutant still retained the ability to bind InsP₆. These results suggest that N-terminal helix α1 is not the major player in InsP₆ binding. Nonetheless, the four InsP₆-contacting residues in the α1 helix of SPX1 are also strictly conserved in SPX1 orthologs (Supplementary Fig. 6a), hinting at an important functional role that is coupled to InsP₆ binding.

**Mechanism of action of InsP₆.** To further dissect the mechanism by which InsP₆ helps disrupt the PHR2 dimer, we superimposed

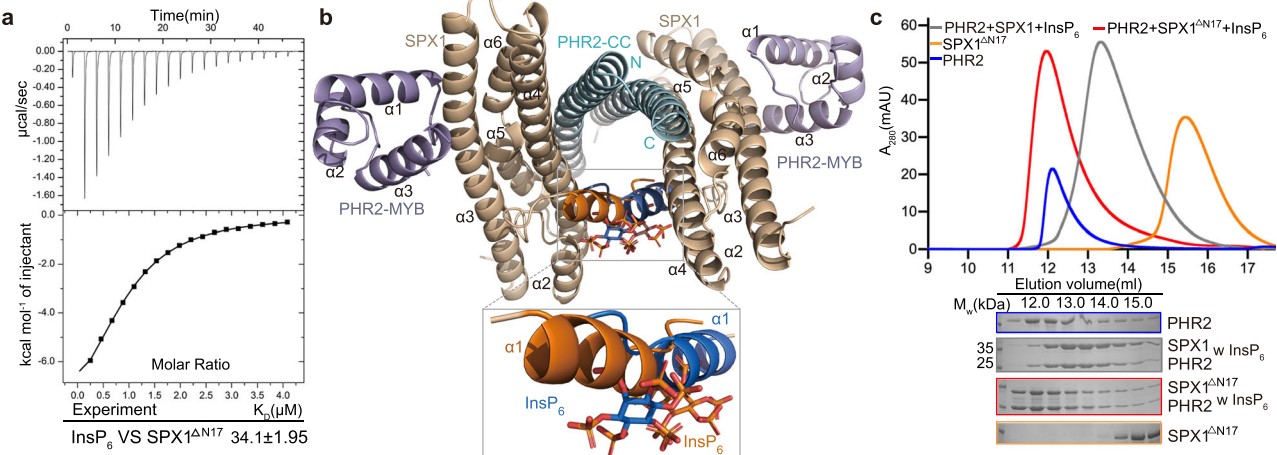

**Fig. 4 SPX1 helix α1 interferes the PHR2 dimer. a** Binding affinity of InsP$_6$ with SPX1$^{1-259\triangle N17}$ (N-terminal helix α1 deletion of SPX1$^{1-259}$) was measured by ITC. **b** PHR2 dimer model constructed by superimposing PHR2 CC in InsP$_6$-SPX1-PHR2 ternary complex structure with AtPHR1 CC dimer structure. The obvious clash occurred between the two helixes α1 was shown in the close-up view. Two helixes α1 of SPX1 were represented in orange and blue respectively. **c** SPX1$^{1-259\triangle N17}$ impaired the SPX1$^{1-259}$-PHR2$^{230-426}$ interaction. SPX1 helix α1 is essential for PHR2 dimer dissociation. The assays were performed as described in (Fig.1a). Experiments were independently repeated three times with similar results. Uncropped gel images are available as source data.

the PHR2 CC motif of our InsP$_6$-SPX1-PHR2 ternary complex structure onto the Arabidopsis PHR1 CC dimer structure. Remarkably, the resulting model readily revealed a severe collision between the α1 helices of the two SPX1 molecules (Fig. 4b). This observation strongly suggests that InsP$_6$ stabilizes SPX1 α1 helix and reinforces a steric hindrance to prevents the SPX1-bound PHR2 to maintain its dimeric state. To validate this idea, we tested the ability of the SPX1 mutant missing its N-terminal α1 helix to disrupt the PHR2 dimer in the presence of InsP$_6$. Although this SPX1 mutant retains its InsP$_6$-binding activity as shown in Fig. 4a, it indeed failed to separate the PHR2 dimer (Fig. 4c and Supplementary Fig. 8). To visualize the effect of InsP$_6$ on the SPX1 α1 helix, we obtained Small Angle X-ray Scattering (SAXS) data for SPX1 and InsP$_6$-SPX1. The comparison of scattering curves between SPX1 and InsP$_6$-bound SPX1 shows that InsP$_6$-bound state appears to be different. It seems that InsP$_6$ makes SPX1 pack tightly according to the smaller Guinier radius of gyration (Rg) and Dmax for SPX1 in presence of InsP$_6$ (Supplementary Fig. 9 and Supplementary Table 2). Further, we calculated the respective molecular bead model using SAXS profiles, and reconstructed the SPX1 model by SASREF. SPX1 and InsP$_6$-bound SPX1 show similar architecture except α1 helix. In addition, the Ab initio molecular envelope shows a difference around α1 helix area (Supplementary Fig. 10). Further, the thermal shift assay showed that InsP$_6$ could increase the melting temperature (Tm) of SPX1 greatly but could not change the Tm of SPX1$^{1-259\triangle N17}$ obviously (Supplementary Fig. 11c, d). The comparison of Circular Dichroism (CD) spectra of SPX1 and InsP$_6$-SPX1 indicated that InsP$_6$ could increase the helical content of SPX1 (Supplementary Fig. 11a, b). Therefore, InsP$_6$ could stabilize SPX1 α1 helix. Notably, the corresponding helix α1 truncation mutation in PHO1, an SPX1 domain-containing protein, exhibited a similar phenotype as SPX1 InsP$_6$-binding site triple mutation[18]. Collectively, the location of the InsP$_6$-binding site distant from the SPX1-PHR2 CC interface, the involvement of the SPX1 α1 helix in InsP$_6$ binding, and the steric hindrance imposed by SPX1 α1 helix in the context of the PHR2 dimer, provide a structural mechanism by which InsP$_6$ allosterically uncouples the PHR2 dimer and promotes the formation of an SPX1-PHR2 1:1 heterodimeric complex.

**Interface between SPX1 and PHR2-MYB**. In addition to breaking the PHR2 dimer apart, our structure shows that SPX1 forms a second interface with PHR2 via its MYB domain. The helices α1 and α3 of the PHR2 MYB domain act as a clamp to hold the helix α3 of SPX1 through a salt-bridge and side-chain hydrogen bond network, which is formed among E89, E92, K99, E100 of SPX1 and K297, H294, R248, E257, R302 of PHR2 (Fig. 5a). To verify the structure, we individually mutated residues in the SPX1-PHR2 MYB interface into alanine. SPX1 mutants E89A, K99A, E92A, and E100A impaired the interaction with PHR2 and PHR2 mutants K297A, H294A, R302A attenuated the binding with SPX1. Other mutants PHR2 R248 and E257 had little effect on the SPX1-PHR2 interaction. These results suggest that this second SPX1-PHR2 interface is also important for SPX1-PHR2 complex assembly (Fig. 5b and Supplementary Fig. 3c, d).

Remarkably, superposition analysis of our structure and the Arabidopsis PHR1 MYB-DNA complex structure reveals a severe steric hindrance for DNA binding by the SPX1-bound PHR2-MYB. In fact, the PHR2 MYB domain contacts with SPX1 through the same helix α3 and Loop1 region, which is used by Arabidopsis PHR1 to dock onto DNA (Fig. 5c). This feature of the InsP$_6$-stabilized SPX1-PHR2 complex provides the structural basis explaining the previous observation that SPX1 competes with DNA for PHR2 MYB binding in vitro and ultimately inhibits the transcriptional activation of PHR2-induced PSI genes. The functional importance of this SPX1-PHR2 MYB interface is further underscored by the conservation of the majority of the interface residues in plants (Supplementary Fig. 6a, b). In fact, Arabidopsis PHR1 mutants K325A/R335A and K325A/H328A/R335A, which have residues at the predicted SPX1-binding interface (R335 and H328 are the PHR1 residues corresponding to rice PHR2 R356 and H349), showed a Pi excess phenotype[20].

## Discussion

Our InsP$_6$-SPX1-PHR2 structure predicates the incompatibility of PHR2 homodimerization and SPX1 binding due to the collision of two SPX1 helices α1 in superimposition analysis and mechanistically explains the ability of SPX1 in disrupting PHR2

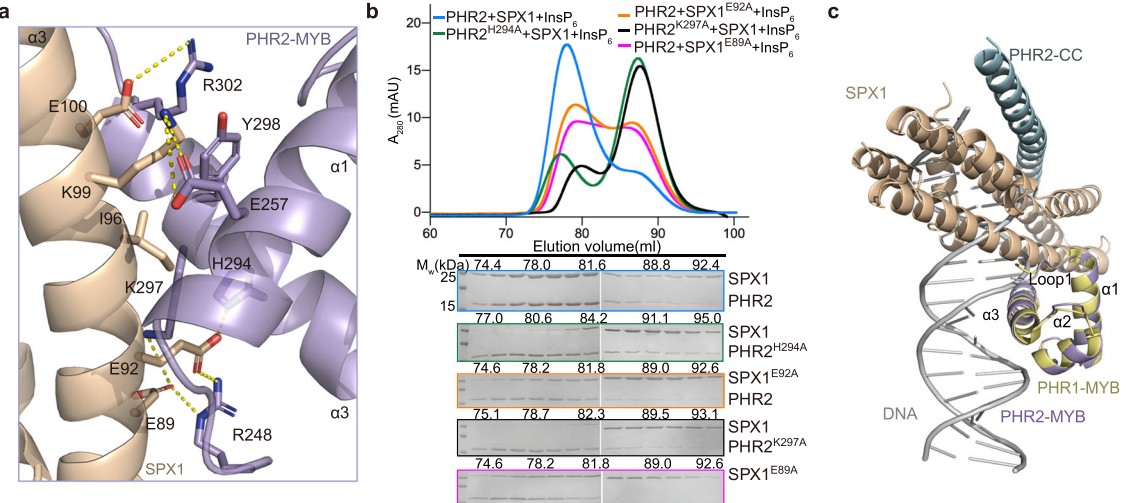

**Fig. 5 Structural basis for PHR2 DNA binding inhibition by SPX1. a** Close-up view of SPX1-PHR2 MYB interface. Critical interface residues were shown as sticks and labeled. The yellow dashed lines represent hydrogen bonds. **b** Mutations disrupted the $SPX1^{1-198}$-$PHR2^{248-380}$ complex formation. (Upper) Gel filtration profiles of $SPX1^{1-198}$ mutants and $PHR2^{248-380}$ mutants were color-coded. (Lower) Coomassie-blue stained SDS-PAGE gels of peak fractions. Experiments were independently repeated three times with similar results. Uncropped gel images are available as source data. **c** SPX1 occupied the PHR2 MYB-DNA binding region. Helix α3 and loop 1 of MYB involve in DNA binding and SPX1 interacting based on superimposition of SPX1-PHR2 complex and AtPHR1 MYB-DNA complex (PDB:6J4R).

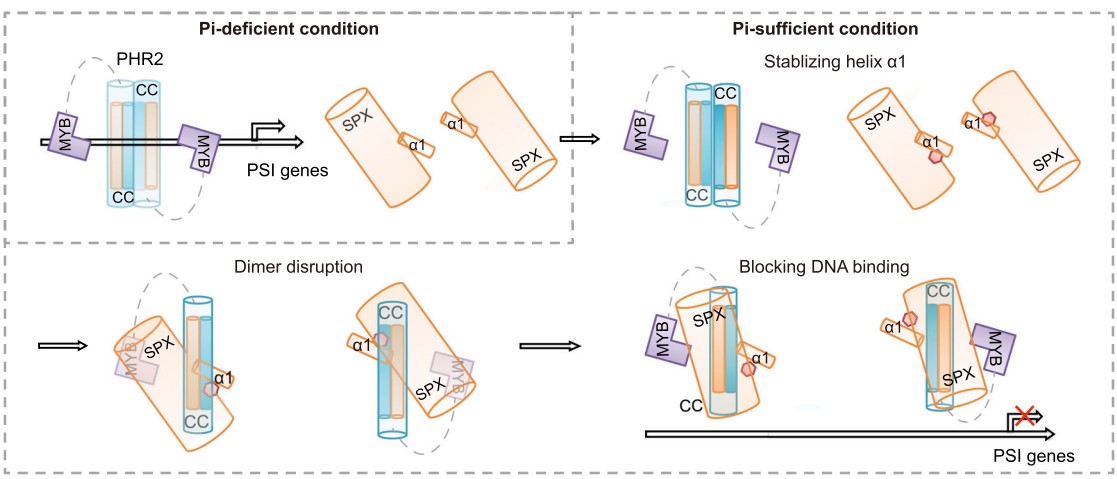

**Fig. 6 InsP$_6$-induced and SPX1-mediated PHR2 dimer dissociation and DNA binding inhibition.** A proposed model for InsP molecule induced recognition and inhibition of PHR2 by SPX1. Under Pi-deficient condition, PHR2 existed as a dimer to bind to the P1BS, and activated the transcription of PSI genes. Under Pi-sufficient condition, InsP molecule could be recognized by SPX1 and stabilized the helix α1 of SPX1 with InsP molecule concentration increasing. Then, PHR2 dimer was disrupted, allowing SPX1 interacted with MYB and CC domains of PHR2 simultaneously. Thus, the transcriptional activation of PSI genes was repressed. The red octagon in the model represented InsP molecule.

dimer in the presence of InsP$_6$. Interestingly, the InsP$_6$ binding site is not situated at the SPX1-PHR2 interface. But the loss of InsP$_6$ binding ability clearly affects the stable interaction between SPX1 and PHR2. Based on these observations, we conclude that InsP$_6$ plays an important role in SPX1-mediated PHR2 dimer dissociation. A similar effect on PHR2 attenuation was observed in SPX1 helix α1, whose deletion did not affect InsP$_6$ binding greatly, but prevented the dimer separation despite of helix α1 absence on the SPX1-PHR2 interaction interface. However, neither InsP$_6$ nor SPX1 itself could abolish the PHR2 dimer (Fig. 1a and Supplementary Fig. 12a, b). Since helix α1 stabilizes SPX1-InsP$_6$ interaction, InsP$_6$ is expected to elicit the same effect on SPX1 helix α1, which leads to the destabilization of the PHR2 dimer. We hypothesize that the InsP molecule is sensed by SPX1 and binds to SPX1 with the increase of InsP concentration. Each

InsP molecule from two SPX1 molecules probably stabilizes SPX1 helix α1 to disrupt PHR2 dimer. Concurrently with the dimer dissociation, PHR2 MYB domain is occupied by SPX1, and its DNA binding is blocked and hence loses the ability to promote expression of PSI genes (Fig. 6a).

Many signal molecules transmit signal to downstream through a common allosteric mechanism such as gibberellin and strigolactone[24,25]. InsP molecules allosterically triggered PHR2 dimer separation, a key step for complex formation, which is reminiscent of Abscisic acid (ABA). ABA could weaken ABA receptor PYL dimer, and monomeric state of ABA receptor such as PYL10 and PYL13 could mediate ABA-independent inhibition of PP2C[26,27]. This is in line with our result that monomeric PHR2 could directly interact with SPX1 independently of InsP molecule. ABA induces conformational changes on the dimer stabilization

CL2 loop of PYL1 and weakens the dimer interface[28]. However, our InsP$_6$-SPX1-PHR2 ternary complex structure and assays indicate that InsP$_6$ may attenuate PHR2 dimer by stabilizing SPX1 helix α1, which is different from the conformational changes induced by ABA. By deciphering the InsP sensing and signaling mechanism of SPX1, our findings shed light on a key process of plant adaptations to variations in nutrient availability and pave the road for engineering phosphate efficiency in crops.

## Methods

**Protein expression and purification**. The truncations and mutants of SPX1 or PHR2 were cloned into the modified pET21a vector (Novagen) providing a tobacco etch virus (TEV) cleavable N-terminal 10xHis-MsyB tag. Sequences for primers are listed in Supplementary Table 3. The proteins were overexpressed in E. coli BL21 (DE3). Cells were grown to OD$_{600nm}$ of 0.6 at 37 °C, then reduced to 16 °C and induced with 0.1 mM isopropyl β-D-1-thiogalactopyranoside for 8–10 h. Cells were harvested by centrifugation for 15 min at 5300 × g and resuspended in **buffer A** (50 mM Tris/HCl pH 8.0, 200 mM NaCl, 20 mM imidazole, 1 mM TCEP). Lysates were cleared by centrifugation at 37,000 × g for 1 h. The supernatant was loaded onto a Ni$^{2+}$ affinity column (GE Healthcare). Then, proteins were eluted with buffer A supplemented with 250 mM imidazole. The proteins were loaded onto an anion-exchange column (Source-15Q, GE Healthcare). The target proteins were separated by increasing the salt concentration from **buffer B** (20 mM Tris/HCl pH 8.0, 100 mM NaCl, 1 mM TCEP) to the same buffer containing 1 M NaCl through a linear gradient. After cleavage by TEV protease at 4 °C for 12 h, the proteins were purified by heparin column with the same buffer used in anion-column purification. The target proteins were further purified by Size Exclusion Chromatography (SEC, Superdex 200 Increase 10/300 GL, GE Healthcare) with the **buffer C** containing 20 mM Tris/HCl pH 8.0, 200 mM NaCl, 1 mM TCEP. The complex of SPX1 and PHR2 were assembled by mixing individually purified equimolar SPX1 and PHR2 with InsP$_6$ (Sigma, catalog #P8810) and incubating at 4 °C for 12 h, then purified by SEC with the **buffer D** containing 200 mM NH$_4$HCO$_3$ pH 8.6, 1 mM TCEP for crystal screening.

**Crystallization, data collection, and structure determination**. The crystals of PHR2 (residues 248–380) and SPX1 (residues 1–198) complex showed obvious anisotropy, and the T4-lysozyme was fused to SPX1 for improving the crystal quality. The complex crystals were grown at 4 °C by the hanging-drop vapor diffusion method, using 1.5 μl protein complex sample mixed with an equal volume of reservoir solution (0.1 mM sodium phosphate dibasic/citric acid, pH 4.2, 200 mM NH$_4$HCO$_3$, 10% (w/v) PEG3350). Furthermore, good quality crystals were subjected to a post-crystallization dehydration procedure by gradually increasing the concentration of PEG3350 to 20%, then directly frozen in liquid nitrogen with cryoprotectant (0.1 mM sodium phosphate dibasic/citric acid pH 4.2, 200 mM NH$_4$HCO$_3$, 18% (w/v) PEG3350, 20% glycerol). These procedures improved the resolution of the crystal to 2.6 Å. PHR2$^{248-380\ V263M/L278M/L295M/L340M}$ mutation, with no influence on SPX1 binding, was used to facilitate structure determination. The phase of the SPX1$^{1-198-T4\ lysozyme}$–Se-PHR2$^{248-380\ V263M/L278M/L295M/L340M}$ complex was calculated by single-wavelength anomalous dispersion method using data collected at the peak wavelength of selenium (λ = 0.979 Å). All data sets were collected at beamlines BL17U1 and BL19U1 at the Shanghai Synchrotron Radiation Facility. The crystals belonged to space group P2$_1$2$_1$2$_1$ with unit cell parameters a = 65.24 Å, b = 107.49 Å, c = 174.52 Å, and contained two SPX1-PHR2 complex molecules in an asymmetric unit. Reflection data were indexed, integrated, and scaled with XDS[29]. The structural model was manually built, refined, and rebuilt using COOT and PHENIX[30,31]. The final refinement statistics are summarized in Supplementary Table 1.

**Gel filtration assay**. The HiLoad 16/600 Superdex 200 pg column and Superdex 200 Increase 10/300 GL column were used for the SEC analysis. The assay was performed with an injection volume of 1 mL, and the **buffer C** (20 mM Tris/HCl pH 8.0, 200 mM NaCl, 1 mM TCEP) with or without InsP$_6$ (Figs. 1b, c, 2c, 4c, 5b, and Supplementary Figs. 3a–d, 12a, b), the **buffer C** or with InsP$_6$/InsP$_8$ (Fig. 3c), and **buffer E** (200 mM PBS pH7.4) or with InsP$_6$ (Fig. 1a). All the samples were incubated at 4 °C for 12 h. The peak fractions were analyzed by SDS-PAGE followed Coomassie Brilliant Blue staining.

**SEC coupled with multi-angle light scattering**. The analysis was performed on an AKTA Pure system (GE Healthcare) coupled with a static light scattering detector (miniDawn, Wyatt) and a differential refractive index detector (Optilab, Wyatt). Protein samples (concentration of 2–3 mg ml$^{-1}$) were filtered and loaded onto a Superdex 200 Increase 10/300 GL column pre-equilibrated by the **buffer C** (20 mM Tris/HCl pH 8.0, 200 mM NaCl, 1 mM TCEP) with or without 1 mM InsP$_6$. Data were analyzed and drawn with ASTRA7 (Wyatt). The experiments were performed at 4 °C.

**Isothermal titration calorimetry**. The Isothermal Titration Calorimetry (ITC) experiments were carried out on a Microcal PEAQ-ITC instrument (Malvern). All titrations were performed at 25 °C. Samples in cell and syringe were both dialyzed against **buffer E** (200 mM PBS pH7.4) (Fig. 1e) or **buffer F** (20 mM HEPES/NaOH pH 7.5, 200 mM NaCl, 2 mM β-mercaptoethanol) (Figs. 1d, 3b, 4a and Supplementary Fig. 7) prior to all titrations. To measure the binding affinity between P1BS (CGCG AATATTCCCA) and PHR2$^{230-426}$, PHR2$^{230-426}$-SPX1$^{1-259}$, PHR2$^{230-426}$-InsP$_6$, PHR2$^{230-426}$-SPX1$^{1-259}$-InsP$_6$ or PHR2$^{230-426\ L348A/L358A/I362A}$, 40 μl P1BS at 70 μM was injected into 280 μL protein samples at 10 μM respectively. To measure the binding affinity between SPX1$^{1-259}$, SPX1$^{1-259\ Y25F/K29A/K151A}$ or SPX1$^{1-259}$ K147A and InsP$_6$, 40 μl InsP$_6$ at 600 μM in the syringe was injected into 280 μL SPX1$^{1-259}$, SPX1$^{1-259Y25F/K29A/K151A}$ and SPX1$^{1-259K147A}$ at 40 μM in the cell, respectively. To measure the binding affinity between SPX1$^{1-259ΔN17}$ and InsP$_6$, 40 μl InsP$_6$ at 1.5 mM in the syringe was injected into 280 μL SPX1$^{1-259ΔN17}$ at 70 μM in the cell. To measure the binding affinity between SPX1$^{1-259}$ and PHR2$^{230-426\ L348A/L358A/I362A}$ in presence or absence of 1 mM InsP$_6$, 40 μl PHR2$^{230-426\ L348A/L358A/I362A}$ at 120 μM was injected into 280 μL SPX1$^{1-259}$ at 10 μM. The ITC data were analyzed using Origin 7.0 and Microcal PEAQ-ITC analysis software.

**Plant materials and growth conditions**. The spx1 spx2 mutant was kindly offered from Dr. Chuanzao Mao (Zhejiang University, China). Hydroponic experiments were conducted using rice culture nutrient solution containing 500 μM KH$_2$PO$_4$. The nutrient solution was adjusted to pH 5.5 by KOH and replaced every 3 days. Experiments were carried out in a greenhouse with 16 h day (30 °C)/8 h night (22 °C) photoperiod and photon density 200 μmol•m$^{-2}$•s$^{-1}$.

**Generation of transgenic plants**. The coding sequence of OsSPX1 was amplified and subcloned into pMD19-T vector. Mutations of InsP binding sites (Y25F/K29A/K151A) in OsSPX1 were generated by PCR. The OsSPX1 and mutated OsSPX1 sequences were cloned into the pCAMBIA2301 vector. The 35S promoters were replaced by the OsSPX1 native promoter. These two constructs were confirmed by sequencing and transformed into spx1 spx2 mutant through Agrobacterium-mediated transformation. Sequences for primers are listed in Supplementary Table 3.

**Quantification of Pi content**. The Pi content of 3-week-old seedlings was measured using the phosphomolybdate colorimetric assay[32]. The shoots were collected after fresh weight measurement and ground into fine powder in liquid nitrogen. Inorganic phosphate (Pi) was extracted in 1% acetic acid by repeated freezing and thawing twice. The supernatant was collected and then mixed with ammonium molybdate and ascorbic acid, in which Pi concentration was measured by colorimetric assay at OD = 820 nm.

**Real-time quantitative PCR**. Total RNA was extracted from root of 3-week-old rice plants using the RNeasy Plant Kit (Qiagen). 1 μg total RNA was used for cDNA synthesis with PrimeScript RT reagent Kit (TaKaRa). Real-time qPCR was performed using a QuantiNova SYBR Green PCR Kit (Qiagen) on CFX96 real-time PCR detection system (Bio-Rad). Relative transcripts level were calculated by 2$^{-ΔΔCt}$ method according to the Ct values. Rice OsACT2 was used as the internal reference. Sequences for primers are listed in Supplementary Table 3.

**Dual-LUC transient transcriptional activation assay**. Protoplast isolation and transactivation assays are based on a previous report[33]. Hydroponic culture system was used for phr1phl1 mutant plants growth as described[34]. phr1phl1 mutant plants were grown in ½ Hoagland nutrient solution for 4 weeks and transferred to Pi-depleted condition for 24 h. About four-week-old plants were used for protoplast isolation. After transferring the plasmids, the protoplast from spx1spx2 mutant plants was incubated under Pi-replete condition for 12 h. The harvested protoplasts were quantified with luciferase assay kit (Promega, USA), and LUC luminescence was measured with a plate reader (Perkin Elmer, USA). β-glucuronidase (GUS) reporter plasmid was used as internal control to normalize transfection efficiency in protoplast assays. Sequences for primers are listed in Supplementary Table 3.

**Crosslinking experiment**. Purified OsPHR2$^{230-426}$ protein was concentrated to 4 mg ml$^{-1}$ in 25 mM HEPES, pH7.0, 150 mM NaCl. About 2 mg of crosslinker DSS (disuccinimidyl suberate) was dissolved in 108 μl DMSO to a final concentration of 50 mM. 2 μl crosslinker DSS was added into 18 μl protein sample at increasing concentration (0, 0.02, 0.04, 0.06, 0.08, and 0.10 mM,) according to the manufacturer's protocol. The reaction was carried out for 30 min at room temperature and stopped by the addition of 50 mM Tris-HCl buffer, pH 8.0. After incubation of the mixture for 15 min at room temperature, the oligomeric state of samples was analyzed by SDS-PAGE electrophoresis.

**Thermal shift assay**. Thermal shift assays were conducted with 15 μM of SPX1$^{1-259}$ or SPX1 $^{1-259\Delta N17}$ in 20 mM HEPES/NaOH pH 7.5, 200 mM NaCl, and 5×dilution of SYPRO Orange dye (S6650, Thermo Fisher Scientific). The concentration of InsP$_6$ ranges from 0 to 250 μM. Protein samples were heated with a 0.5 °C per 10 s increasing gradient from 10 to 95 °C, the melt curve and melt temperature were recorded by the CFX Connect Real-Time PCR detection system (Bio-Rad).

**Circular dichroism measurements**. The CD spectra of the protein samples were recorded by a Chirascan™-plus CD Spectrometer at the room temperature. The protein samples were concentrated to 25 μM in 20 mM HEPES/NaOH pH 7.5, 300 mM NaF in the absence or presence of 1 mM InsP$_6$.

**SAXS data collection and analysis**. The small angle X-ray scattering data were collected at the BL19U2 beamline at National Facility for Protein Science Shanghai (NCPSS) and Shanghai Synchrotron Radiation Facility (SSRF). 60 μL of SPX1$^{1-259}$ 1.8 mg/mL, InsP$_6$-bound SPX1$^{1-259}$ 1.1 mg/mL were loaded in a quartz capillary. 2D scattering images were converted to 1D SAXS curves by BioXTAS RAW. The matching buffer scattering was subtracted from the sample scattering by PRIMUS. Pair distribution functions of the particles P(r) and the maximum sizes Dmax were calculated by the program GNOM. Low-resolution shapes were determined from solution scattering data using DAMMIF, from the ATSAS suite of programs. Twenty independent calculations were performed by DAMMIF programs for each data set, using default parameters and no symmetry constraints. Then twenty independent reconstructions were then averaged and filtered to a final consensus model using the DAMAVER suite. Rigid body modeling was performed using the program SASREF. We used SASREF to find relative positions of the helix α1 and the remaining part of SPX1 by inputting both models separately.

**Reporting summary**. Further information on research design is available in the Nature Research Reporting Summary linked to this article.

## Data availability

Structural coordinates and structural factors have been deposited in the Protein Data Bank under accession number 7E40. Source data are provided with this paper.

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

## Acknowledgements

We thank Prof. Ning Zheng for critically reading our manuscript and for giving detailed comments and suggestions; the staff members of the BL17U1, BL19U1, and BL19U2 beamlines of National Facility for Protein Science in Shanghai (NFPS) at Shanghai Synchrotron Radiation Facility (SSRF) in China for support during crystallographic data and small angle X-ray scattering data collection; the staff members of the Large-scale Protein Preparation System at the National Facility for Protein Science in Shanghai (NFPS), Zhangjiang Lab, China for SEC-MALS data collection and analysis. M.L. is supported by Chinese Academy of Sciences and National Natural Science Foundation of China (31970580).

## Author contributions

Experimental design: J.Z., Q.H., and W.X.; different constructs for protein purification, the complex assembling, crystallization, and crystal optimization: J.Z. and Q.H.; attempt in different species' protein for crystallization: H.L. and R.L.; structure determinations: D.Y.; structure analysis and various mutations design: J.Z., Q.H., and W.X; SEC assays: J.Z.; SEC-MALS and ITC assays: J.Z. and Q.H.; circular dichroism measurements: J.Z., J.Y., and C.W.; plant experiments design: J.Z., J.-K.Z., M.L., and W.X.; plant experiments: X.X. and S.G.; SAXS data collection and analysis: J.Z. and R.C.; thermal shift assay: J.Z. and Q.H.; manuscript writing: J.Z., Q.H., F.M., M.L., and W.X. with help from all of the co-authors.

## Competing interests

The authors declare no competing interests.
