## [Peer Review File · Nature Communications]

Mechanism of Phosphate Sensing and Signaling Revealed by Rice SPX1-PHR2 Complex StructureREVIEWER COMMENTS

Reviewer #1 (Remarks to the Author):

The manuscript by Zhou, et al. presents a new structure and interaction data with some genetics suggesting how InsP6 mediates biochemical interaction between SPX1 and PHR2. The InsP6-mediated alpha 1 helix stabilization mechanism proposed is very interesting, suggesting the interaction of SPX helix 1 with InsP6 interferes with PHR oligomerization. The new structure and mechanism warrant publication, and overall has potential to be a nice contribution. However, the authors mis-characterize the extent to which InsP6 is required for SPX-PHR biochemical interaction throughout the manuscript and some clarification of experimental details is required. Thus major revisions are required.

Major criticisms:

1. Conclusions from Fig 1 are overstated. The gels in Fig 1A show about half the SPX1 co-migrates with PHR2 in the absence of InsP6, but the authors state "SPX1 and PHR2 proteins co-migrated on the column only in the presence of InsP6...". InsP6 generates a new peak, and the crystal structure defines what's in that peak, but it is not clear enough what's in the multiple SPX-PHR-alone peaks to say NONE of them is an SPX-PHR dimer, perhaps in a frustrated alternative conformation. The authors can either focus on the known dimer and their crystal structure describing it, or provide additional orthogonal evidence defining the two states (with and without InsP6) if they want to talk about InsP6 induction of the dimerized state.
2. Given the dynamics of the Alpha 1 helix mechanism suggested by the authors, some kind of in-solution dynamics study would provide orthogonal evidence that InsP6 stabilizes the alpha 1 helix, e.g. HDX, HSQC, SAXS, SANS, crosslinking as in Supp Fig 2A, engineered disulfide trapping or even just CD, limited protease protection or melting temp differential scanning fluorimetry. Any of these studies might also provide additional evidence on the structure / dimerization state of the InsP6-free complex mentioned above in #1.
3. Although the genetics are convincing in Fig 3D-G, the SEC in Fig 3C are confusing. In Fig 1A the amounts of SPX and PHR loaded appear equal, but in Fig 3C there appears to be less PHR loaded on the SEC by OD and in the gels. The SPX mutant has a Y25F change which could (maybe) increase OD a little, but still doesn't fully explain the increased SPX OD. The PHR OD should be the same in Fig 1 and Fig 3. Were equimolar amounts of SPX and PHR loaded? If so why the different PHR OD? If different amounts of protein were used the SEC should be re-run with equal molarities of PHR and SPX mutant. Further, since SPX mutant loses function with InsP6, it would be easy to use Ins7P or Ins8P species to validate the result on different higher order InsP species, addressing two criticisms in one experiment.

Less Major:

4. Rama favored & allowed were in methods but the authors did not mention the 2 outliers in Chain D for some reason. All Ramachandran stats, including the two outliers, should be included in the crystallography table. And the outliers for A61 and R19 explained in the main text. There were also too many real space outliers so the authors should at least explain their attempts to fit those and/or why they had trouble doing so.

5. Methods say two different buffers were used for the ITC, but no mention if they were matched. Authors make statements about the K_d 's while the baseline energies are different. This just needs clarification that buffers were matched, but error for the K_d 's need to be reported in order to say K_d 's are same or different.

6. Line 173. The authors tested interface residues for SPX-PHR complex formation, but since all the experiments had InsP6 present, they did not test Ins6P dependence. Authors just need to cut out the "InsP6-induced" and then the statement is at least accurate.

7. Line 169, authors state L360A contributes to interface formation based on the crystal structure, but when they tested this residue in Supp Fig 5B, the L360A mutant does not affect interaction (note "the most orange" line in Supp Fig 5B). This phenomenon should be clearly acknowledged in the main text for clarity.

Minor:

8. Line 243, Authors say again the tested MYB interface is important for InsP6-induced interactions but they never test it. Just remove "Insp6-induced" and at least the statement will be accurate.

9. Discrepancies in conclusions throughout the manuscript, e.g. line 682 vs.169. Line 85 vs. line 107. Line 107 vs. Lines 111-13.

10. Fig Supp 2A, Line 637, figure legends should describe the experiment, not repeat conclusions.

11. Throughout the paper, line colors in figures cannot be distinguished (such as dark blue vs. darker blue vs. blue). Perhaps also use dashes/dots to distinguish?

12. Line 529, typo in listed buffers.

13. Fig S1B-C, Lysozyme is misspelled.

14. Fig 3B, ITC needs error for the reported K_d 's.

15. Was there a reason K147 was not mutated in SPX1 for the InsP6 binding site? No worries either way, but if that mutant was made it should be reported.

16. Legends are carelessly labeled with little attention to detail.

17. Lines 220-222 are difficult to understand.

Reviewer #2 (Remarks to the Author):

The PHR family transcription factors are the key players in Pi signaling. In the presence of InsP molecules, SPX family proteins can interact with PHR proteins, inhibiting their gene activation ability and avoiding the toxicity of high concentration of Pi. In this manuscript, Zhou et al. determined the crystal structure of InsP6-SPX1-PHR2 complex, revealed the detailed interaction between SPX1 and PHR2 CC domain. Instead of SPX1-PHR2 interacting interface, in vitro mutagenesis and gel filtration assays confirmed that InsP6 binds to SPX1 and disrupts the dimerization of PHR2 via the steric hindrance of SPX1 N-terminal helices. Unlike the apo-form PHR2, the MYB domain of PHR2 is incompatible with dsDNA binding upon InsP6-SPX1-PHR2 complex formation. All experiments were well designed and performed. This work advanced our understanding of the molecular mechanism how SPX1 inhibits PHR2 from binding to DNA upon sensing Pi signals.

Concerns:

- 1) Fig. 3a, please check the labels for the right panel. Are the labels K5 and K2 switched?
- 2) According to Table S1, the B-factors of InsP6 are higher than those of protein and water molecules. What is the occupancy of InsP6? What are the possible causes of the high B-factors? Instead of 2Fo-Fc electron density map, please provide simulated annealing omit map for InsP6 and InsP6-interacting residues. In addition, please include the Ramachandran parameter in Table S1.
- 3) MYB domain-containing proteins usually bind to dsDNAs. The sequence of P1BS (CGCGAATATTCCCA) is not perfectly self-complementary. Is the complementary strand of P1BS present in the ITC assay?
- 4) In addition to gel filtration assay (Fig. 4c), it will be helpful to confirm the oligomerization states of SPX1Δ17-PHR2-InsP6 complex by other methods, such as SEC-MALS assay used for other complex.
- 5) As depicted in Fig. S3b low panel, there are certain conformational difference between SPX-bound and CtGde1-bound InsP6. Are the InsP6-interacting residues conserved in these two proteins?

Comments and concerns from Reviewer 1

The manuscript by Zhou, et al. presents a new structure and interaction data with some genetics suggesting how InsP6 mediates biochemical interaction between SPX1 and PHR2. The InsP6-mediated alpha 1 helix stabilization mechanism proposed is very interesting, suggesting the interaction of SPX helix 1 with InsP6 interferes with PHR oligomerization. The new structure and mechanism warrant publication, and overall has potential to be a nice contribution. However, the authors mis-characterize the extent to which InsP6 is required for SPX-PHR biochemical interaction throughout the manuscript and some clarification of experimental details is required. Thus major revisions are required.

Major criticisms 1: *Conclusions from Fig 1 are overstated. The gels in Fig 1A show about half the SPX1 co-migrates with PHR2 in the absence of InsP6, but the authors state “SPX1 and PHR2 proteins co-migrated on the column only in the presence of InsP6...”. InsP6 generates a new peak, and the crystal structure defines what’s in that peak, but it is not clear enough what’s in the multiple SPX-PHR-alone peaks to say NONE of them is an SPX-PHR dimer, perhaps in a frustrated alternative conformation. The authors can either focus on the known dimer and their crystal structure describing it, or provide additional orthogonal evidence defining the two states (with and without InsP6) if they want to talk about InsP6 induction of the dimerized state:*

Response: We agree with this comment. It is noticed that there is a positive charged patch in PHR2 and negative charged patches in SPX1. Additionally, we found PHR2 could bind to an acidic protein MsyB (a tag) by gel filtration assay. The PHR-MsyB interaction should be mediated by electrostatic force and be not related to their function. It is highly possible that PHR2 could nonspecifically interact with negative charged proteins. We noticed that SPX1 could nonspecifically interact PHR2 under low salt. When we increased ionic strength by changing buffer from 200mM NaCl to 200mM PBS, the non-specific interaction is much weaker. Further, as shown in Figure 1e, the possible nonspecific (dynamic) interaction between SPX1 and PHR2 could not inhibit DNA binding ability of PHR2. We believe the interaction should be dynamic and unspecific without InsP₆, and is not related to their function. So, we changed the buffer (increase the ionic strength) and rerun this assay to alleviate the non-specific interaction. We replaced Figure 1a with a new figure to avoid misunderstanding. On the other hand, we emphasized the peak shift of PHR2 dimer in presence of InsP₆, and removed “only” (line 86-91). We removed gel filtration data for MBP-tagged or H2A-tagged PHR2²⁴⁸⁻³⁸⁰ and SPX1¹⁻¹⁹⁸ since peak shift could be observed when fusion tag existed (line 109-113).

Major criticisms 2: *Given the dynamics of the Alpha 1 helix mechanism suggested by the authors, some kind of in-solution dynamics study would provide orthogonal evidence that InsP6 stabilizes the alpha 1 helix, e.g. HDX, HSQC, SAXS, SANS, crosslinking as in Supp Fig 2A, engineered disulfide trapping or even just CD, limited protease protection or melting temp differential scanning fluorimetry. Any of these studies*

might also provide additional evidence on the structure / dimerization state of the InsP6-free complex mentioned above in #1

Response: We really appreciate the suggestion. To visualize the conformational changes of SPX1 induced by InsP₆, we obtained SAXS data for SPX1 and InsP₆-SPX1. The comparison of scattering curve between SPX1 and InsP₆-bound SPX1 shows that InsP₆-bound state appears to be different via SAXS. Based on SAXS analysis, the Guinier radius of gyration (R_g) and D_{max} for SPX1 is ~28.03 Å and 100 Å, larger than 25.33 Å and 89 Å of InsP₆-bound SPX, calculated from their SAXS profiles (Supplementary Fig.10). This result suggests that SPX1 is less compact than InsP₆-bound SPX1. Further, we calculated respective molecular bead model using SAXS profiles, and reconstructed the SPX1 model by SASREF (Supplementary Fig.11). SPX1 and InsP₆-bound SPX1 show similar architecture except α 1 helix. In addition, the Ab initio molecular envelope show difference around helix α 1 area. Meanwhile, we performed thermal shift assay and compared Circular Dichroism (CD) spectra of SPX1 and InsP₆-SPX1 (Supplementary Fig.12). The thermal shift assay showed InsP₆ could stabilize SPX1 greatly but could not change the stability of SPX1 ^{Δ N17} prominently. InsP₆ could increase the content of α helix of SPX1 by the difference of CD spectra of SPX1 and InsP₆-SPX1. We added these experiments and assays in our main text (line246-259) to support the function of InsP₆.

Major criticisms 3: *Although the genetics are convincing in Fig 3D-G, the SEC in Fig 3C are confusing. In Fig 1A the amounts of SPX and PHR loaded appear equal, but in Fig 3C there appears to be less PHR loaded on the SEC by OD and in the gels. The SPX mutant has a Y25F change which could (maybe) increase OD a little, but still doesn't full explain the increased SPX OD. The PHR OD should be the same in Fig 1 and Fig 3. Were equimolar amounts of SPX and PHR loaded? If so why the different PHR OD? If different amounts of protein were used the SEC should be re-run with equal molarities of PHR and SPX mutant. Further, since SPX mutant loses function with Ins6P, it would be easy to use Ins7P or Ins8P species to validate the result on different higher order InsP species, addressing two criticisms in one experiment*

Response: Thank you for pointing this out. We noticed that we did use less PHR2 than SPX1 in Figure 3c. We rerun this assay using equal amounts of SPX1 and PHR2, which is also similar to protein loading in Figure 1a. We replaced Figure 3c with a new figure. Regarding to higher order InsP species, we used InsP₈ to validate the function of SPX1 mutant. We found that neither InsP₆ or InsP₈ could promote the interaction between PHR2 and SPX1 mutant. So, InsP₈ probably binds to similar position of SPX1 to promote two protein interaction (line 216-219).

Less Major criticisms 4: *Rama favored & allowed were in methods but the authors did not mention the 2 outliers in Chain D for some reason. All Ramachandran stats, including the two outliers, should be included in the crystallography table. And the outliers for A61 and R19 explained in the main text. There*

were also too many real space outliers so the authors should at least explain their attempts to fit those and/or why they had trouble doing so.

Response: Thank you for pointing this out. We use <https://servicesn.mbi.ucla.edu/PROCHECK/> server to calculate Ramachandran data. The result is: 93.8% core 5.9% allow 0.4% gener 0.0% disall. Now we noticed it is different from the result calculated by PDB validation. Now we updated our Ramachandran data and mentioned these two outliers (located in the loop region) (line 147-149). For several real space outliers, we have tried several times but failed to fix it. The probably reason for these outliers is that the quality of density is not high enough to restrain.

Less Major criticisms 5: *Methods say two different buffers were used for the ITC, but no mention if they were matched. Authors make statements about the Kd 's while the baseline energies are different. This just needs clarification that buffers were matched, but error for the Kd's need to be reported in order to say Kd's are same or different.*

Response: Thank you for pointing this out. We used buffer F (20 mM HEPES/NaOH pH 7.5, 200 mM NaCl, 2 mM β -mercaptoethanol) for all ITC assay except the assay in Figure 1e (buffer E (200 mM PBS pH7.4) is used). These two buffers are both suitable for ITC assay. As we mentioned in our response to Major Criticisms 1 that 200 mM PBS could alleviate most of nonspecific SPX1-PHR2 interaction, it is more accurate to use buffer E (200 mM PBS pH7.4) than buffer F in ITC assay for evaluating the effect of SPX1 on PHR2 DNA binding. Actually, even using buffer F, SPX1 itself hardly affect PHR2 DNA binding, similar to the result using buffer E. For the difference of baseline energies, we realized that the InsP₆ for our previous assay is made by resolving InsP₆ powder into Tris buffer, which is not a suitable buffer for ITC assay. Now we made InsP₆ solution using water and pH was adjusted to 8.0 by NaOH. Now baseline energies are almost the same using new InsP₆ solution. So, we replaced Figure 1d, Figure 1e, Figure 3b, Figure 4a with new figures. In addition, the errors for the reported Kd's have been put in the results.

Less Major criticisms 6: *Line 173. The authors tested interface residues for SPX-PHR complex formation, but since all the experiments had InsP6 present, they did not test Ins6P dependence. Authors just need to cut out the "InsP6-induced" and then the statement is at least accurate.*

Response: Thank you for pointing this out. "InsP₆-induced" has been deleted (line 188).

Less Major criticisms 7: *Line 169, authors state L360A contributes to interface formation based on the crystal structure, but when they tested this residue in Supp Fig 5B, the L360A mutant does not affect interaction (note "the most orange" line in Supp Fig 5B). This phenomenon should be clearly acknowledged in the main text for clarity.*

Response: Thank you for pointing this out. When we analyze the structure, The L360 of PHR2 looks like participate the hydrophobic patch formation, however, PHR2 single mutation L360A retained the ability to

bind SPX1. Thus, PHR2 L360A does not play the major role in SPX1 interaction. We forgot to remove L360. Now, we mentioned the result of L360 in the maintext (line 181-186).

Minor criticisms 8: *Line 243, Authors say again the tested MYB interface is important for InsP6-induced interactions but they never test it. Just remove “Insp6-induced” and at least the statement will be accurate.*

Response: Thank you for pointing this out. “InsP₆-induced” has been deleted (line 277).

Minor criticisms 9 *Discrepancies in conclusions throughout the manuscript, e.g. line 682 vs.169. Line 85 vs. line 107. Line 107 vs. Lines 111-13*

Response: Thank you for pointing these discrepancies out. For the discrepancy line 682 vs.169, we have corrected as we mentioned in our response to Less Major criticisms 7 (line 181-186). For the discrepancy line85 vs. line 107, we rerun this assay using new buffer to alleviate nonspecific interaction (Figure 1a). We removed “only” as our response to Major criticisms 1 (line 87). For the discrepancy Line 107 vs. Lines 111-13, as we mentioned in our response to Major criticisms 1, SPX1 and PHR2 form stable, specific interaction under InsP₆. The interaction should be dynamic and unspecific without InsP₆. When we increase the ion strength in buffer, the non-specific interaction is much weaker. So, we claimed that if stable binding of SPX1 to PHR2 hinges on PHR2 monomerization.

Minor criticisms 10: *Fig Supp 2A, Line 637, figure legends should describe the experiment, not repeat conclusions.*

Response: Thank you for pointing this out. We have corrected it by explaining the experiment (line 726-729).

Minor criticisms 11: *Throughout the paper, line colors in figures cannot be distinguished (such as dark blue vs. darker blue vs. blue). Perhaps also use dashes/dots to distinguish?*

Response: Thank you for the suggestion. We have adjusted the colors of the lines with easily distinguishable colors.

Minor criticisms 12: *Line 529, typo in listed buffers.*

Response: Thank you for pointing out the typos error. It has been corrected (line 577).

Minor criticisms 13: *Fig S1B-C, Lysozyme is misspelled.*

Response: Thank you for pointing out the typos error. It has been corrected.

Minor criticisms 14: *Fig 3B, ITC needs error for the reported Kd's.*

Response: Thank you for pointing this out. The errors for the reported Kd's have been put in Figure 3b.

Minor criticisms 15: *Was there a reason K147 was not mutated in SPX1 for the InsP6 binding site? No worries either way, but if that mutant was made it should be reported.*

Response: Thank you for pointing this out. Actually, we checked the binding affinity between InsP₆ and SPX1(K147A) using ITC assay. However, mutation K147A had no effect on InsP₆ binding affinity. Now, it is mentioned in main text (line 209-210, Supplementary fig. 8).

Minor criticisms 16: *Legends are carelessly labeled with little attention to detail.*

Response: Thank you for pointing this out. Several labels have been corrected.

Minor criticisms 17: *Lines 220-222 are difficult to understand.*

Response: Thank you for pointing this out. The sentence has been rewritten (line 239-243).

Comments from Reviewer 2

Comments:

The PHR family transcription factors are the key players in Pi signaling. In the presence of InsP molecules, SPX family proteins can interact with PHR proteins, inhibiting their gene activation ability and avoiding the toxicity of high concentration of Pi. In this manuscript, Zhou et al. determined the crystal structure of InsP₆-SPX1-PHR2 complex, revealed the detailed interaction between SPX1 and PHR2 CC domain. Instead of SPX1-PHR2 interacting interface, in vitro mutagenesis and gel filtration assays confirmed that InsP₆ binds to SPX1 and disrupts the dimerization of PHR2 via the steric hindrance of SPX1 N-terminal helices. Unlike the apo-form PHR2, the MYB domain of PHR2 is incompatible with dsDNA binding upon InsP₆-SPX1-PHR2 complex formation. All experiments were well designed and performed. This work advanced our understanding of the molecular mechanism how SPX1 inhibits PHR2 from binding to DNA upon sensing Pi signals.

Concerns 1: *Fig. 3a, please check the labels for the right panel. Are the labels K5 and K2 switched?*

Response: Thank you for pointing out the label errors. The label errors have been corrected.

Concerns 2: *According to Table S1, the B-factors of InsP6 are higher than those of protein and water molecules. What is the occupancy of InsP6? What are the possible causes of the high B-factors? Instead of 2Fo-Fc electron density map, please provide simulated annealing omit map for InsP6 and InsP6-interacting residues. In addition, please include the Ramachandran parameter in Table S1.*

Response: Thank you for pointing this out. The occupancy of InsP₆ is 1.0. It is possible that the density around InsP₆ is not good enough to precisely define the position of InsP₆. As a result, the B-factors of InsP₆ are high. Thank you for the suggestion. We have replaced 2Fo-Fc electron density map with

simulated annealing omit map (Figure 3a, supplementary Fig. 7) and put Ramachandran parameter in supplementary Table 1.

Concerns 3: *MYB domain-containing proteins usually bind to dsDNAs. The sequence of PIBS (CGCGAATATTCCCA) is not perfectly self-complementary. Is the complementary strand of PIBS present in the ITC assay?*

Response: Thank you for pointing this out. PHR2 transcription factor, a central regulator of phosphate signaling, binds to the imperfect palindromic sequence (GNATATNC). According to Arabidopsis PHR1 and DNA structure¹, (GNATATNC) the 5 blue colored deoxynucleotides are responsible for DNA binding specificity. It is expected that palindromic sequence will not affect the binding affinity. Therefore, we did not try the self-complementary sequence in our ITC assay.

Concerns 4: *In addition to gel filtration assay (Fig. 4c), it will be helpful to confirm the oligomerization states of SPX1 Δ 17-PHR2-InsP₆ complex by other methods, such as SEC-MALS assay used for other complex.*

Response: Thank you for the suggestion. Actually, SEC-MALS analysis indicated that SPX1 Δ 17-PHR2 complex formed by incubation SPX1 Δ 17, PHR2 and InsP₆ together was eluted with a molecular mass of about 92.2 kDa (supplementary Fig. 9), consistent with a 2:2 SPX1 Δ 17-PHR2 complex (96.8kDa). Therefore, it will provide further evidence for important roles of α 1 helix in dimer dissociation.

Concerns 5: *As depicted in Fig. S3b low panel, there are certain conformational difference between SPX-bound and CtGde1-bound InsP₆. Are the InsP₆-interacting residues conserved in these two proteins?*

Response: Thank you for pointing this out. Most InsP₆-binding residues are conserved. Residues M1, K2, F3, K5, Y25, K29, K147 and K151 in SPX1 participate in the interaction with InsP₆. The corresponding residues in CtGde1 also contribute to InsP₆-CtGde1 binding. In addition to these residues, 4 more lysine (K30, K128, K132 and K133) from CtGde1 also stabilize InsP₆-CtGde1 interaction. Superimposition shows that the InsP₆-contacting helices are not totally fit, which may be the reason for certain conformational difference between SPX-bound and CtGde1-bound InsP₆.

Reference

1. Jiang, M. *et al.* Structural basis for the Target DNA recognition and binding by the MYB domain of phosphate starvation response 1. *FEBS J* **286**, 2809-2821, doi:10.1111/febs.14846 (2019).

REVIEWERS' COMMENTS

Reviewer #1 (Remarks to the Author):

The authors have done an outstanding job improving this manuscript. barring small improvements there is now outstanding evidence supporting the authors claims. Only two minor corrections need to be added:

A) Authors still do not say ITC syringe vs. sample cell buffers were matched. This must be stated or ITC should not be published.

B) Edits make it unclear if some critical statement have been revised, the old AND new line numbers should be provided showing where old statements were cut out or replaced with new statements.

Previous critiques:

1) Addressed – changed salt to decrease InsP6-free peaks and changed language.

2) Addressed but addition of Chi-square for SAXS data to formally compare datasets +/- InsP6 would be an improvement. Right now its just a qualitative comparison, but a formal, quantitative comparison would be very easy to do. Not required from me but would make it better.

3) Addressed – fixed very sloppy stoichiometry.

4) Addressed – fixed crystallography table and explained real space problems and rama outliers.

5) Authors still do not state syringe and sample cell buffers were MATCHED although other problems with ITC were fixed. What's being injected via syringe must be identical to buffer in the sample cell – I'm sure they did that but they have to state that is the case. Otherwise the ITC data should not be published.

6) Addressed – misleading language removed.

7) Addressed – mistake fixed.

8) Addressed – misleading language removed.

9) Cannot determine – The old AND new line numbers are needed to determine if new statements have been revised.

10-17 were all addressed.

Reviewer #2 (Remarks to the Author):

The authors have addressed all my concerns.

REVIEWERS' COMMENTS

Reviewer #1 (Remarks to the Author):

The authors have done an outstanding job improving this manuscript. barring small improvements there is now outstanding evidence supporting the authors claims. Only two minor corrections need to be added:

A) Authors still do not say ITC syringe vs. sample cell buffers were matched. This must be stated or ITC should not be published.

Response: Thank you for pointing this out. For each ITC assay, the buffer in syringe and sample cell are the same. Now, we mentioned that ‘Samples in cell and syringe were both dialyzed against buffer’ in the ITC assay method (lines 358-359).

B) Edits make it unclear if some critical statement have been revised, the old AND new line numbers should be provided showing where old statements were cut out or replaced with new statements.

Response:

For the discrepancy line 682 vs.169 in the first version of the manuscript (shown in left), it has been revised as shown in the right (line 158-162 in current version). When we analyze the structure, the L360 of PHR2 looks like participate the hydrophobic patch formation, however, PHR2 single mutation L360A retained the ability to bind SPX1. Thus, PHR2 L360A does not play the major role in SPX1 interaction.

682 formation. PHR2 ^{R248A} , PHR2 ^{R257A} , PHR2 ^{E353A} , PHR2 ^{E363A} , and PHR2 ^{L360A} had little effect	→	and L352, L342 and L360 of PHR2 also contribute to the interface formation (Fig. 2b).
683 on the SPX1-PHR2 complex formation. The assays were performed as described in		Substitution of V125 and V132 in SPX1 and L342 and L352 in PHR2 with alanine
684 (Supplementary Fig.2c).		abolished SPX1-PHR2 interaction (Supplementary Fig. 5a, b). The L360 of PHR2
168 hydrophobic residues, such as V125, V132, L169, and F174 in SPX1 and L352, L342 and		looks like participate the hydrophobic patch formation, however, PHR2 single mutation
169 L360 of PHR2 also contribute to the interface formation (Fig. 2b). Substitution of V125	L360A retained the ability to bind SPX1. Thus, Collectively, SPX1-PHR2 CC interface,	
170 and V132 in SPX1 and L342 and L352 in PHR2 with alanine abolished SPX1-PHR2		
171 interaction (Supplementary Fig. 5a, b). Thus, SPX1-PHR2 CC interface, which is largely		

For the discrepancy line85 vs. line 107 in the first version of the manuscript (shown in left), we believe that the weak interaction between SPX1-PHR2 should be dynamic and unspecific in absence of InsP₆, and is not related to their function (we have elaborated the conclusion in our previous response to Major criticisms 1). Now, we emphasized the peak shift of PHR2 dimer in presence of InsP₆, and removed “only” as shown in right (line lines 75-78, lines 95-96 in current version).

84 PHR2 ²³⁰⁻⁴²⁶ interaction. As shown in Fig. 1a, the SPX1¹⁻²⁵⁹ and PHR2²³⁰⁻⁴²⁶ proteins co-	→	characterize the SPX1 ¹⁻²⁵⁹ -PHR2 ²³⁰⁻⁴²⁶ interaction. As shown in Fig. 1a, the SPX1¹⁻²⁵⁹
85 migrated on the column only in presence of InsP₆, suggestive of InsP₆-induced interaction.		and PHR2²³⁰⁻⁴²⁶ proteins co-migrated on the column only in presence of InsP₆,
86 Interestingly, the elution volume of PHR2 shifted from 12.2 ml to 13.2 ml when forming a		suggestive of InsP₆-induced interaction. In contrast to no peak shift of PHR2 in absence
107 (Supplementary Fig. 2 c, d). Interestingly, in the absence of InsP₆, SPX1 was shifted by		of InsP₆, the peak of PHR2²³⁰⁻⁴²⁶ proteins on the column shifted later and co-migrated
108 PHR2 as detected by size exclusion chromatography (Fig. 1a). However, it appeared to	with SPX1¹⁻²⁵⁹ proteins in presence of InsP₆, indicative of InsP₆-induced specific	
	interaction (Fig. 1a). Interestingly, the elution volume of PHR2 shifted from 12.2 ml to	
	interestingly, in the absence of InsP₆, it is noticed that SPX1 was slightly shifted by	
	PHR2 as detected by size exclusion chromatography in the absence of InsP₆ (Fig. 1a).	

For the discrepancy Line 107 vs. Lines 111-13 in the first version of the manuscript (shown in left), we think they are not contradictory any more since we believe that SPX1 dynamically and unspecifically interacts with PHR2 in absence of InsP₆. The specific and stable interaction between SPX1 and PHR2 relies on InsP₆. We deleted “Interestingly” and mentioned the phenomena using sentence: ‘it is noticed that’ (lines 95-101 in current version).

107 (Supplementary Fig. 2 c, d). Interestingly, in the absence of InsP₆, SPX1 was shifted by 108 PHR2 as detected by size exclusion chromatography (Fig. 1a). However, it appeared to 109 dynamically interact with the transcription factor and failed to form a stable and 110 homogenous 1:1 heterodimer with PHR2. 111 If stable binding of SPX1 to PHR2 hinges on PHR2 monomerization, a 112 dimerization-defective mutant of PHR2 should be able to form a complex with SPX1 even 113 in the absence of InsP₆. Based on a recently reported Arabidopsis PHR1 CC dimer structure	➔	Interestingly, in the absence of InsP₆; it is noticed that SPX1 was slightly shifted by PHR2 as detected by size exclusion chromatography in the absence of InsP₆ (Fig. 1a). However, it appeared to dynamically interact with the transcription factor but failed to form a stable and homogenous 1:1 heterodimer with PHR2.⁴ If stable binding of SPX1 to PHR2 hinges on PHR2 monomerization, a dimerization-defective mutant of PHR2 should be able to form a complex with SPX1 even in the absence of InsP₆. Based on a recently reported Arabidopsis PHR1 CC dimer
---	---	---

Previous critiques:

1) Addressed – changed salt to decrease InsP₆-free peaks and changed language.

2) Addressed but addition of Chi-square for SAXS data to formally compare datasets +/- InsP₆ would be an improvement. Right now its just a qualitative comparison, but a formal, quantitative comparison would be very easy to do. Not required from me but would make it better.

Response: We performed quantitative comparisons of datasets +/- InsP₆ as shown in Supplementary Figure 9. The R_g and D_{max} for SPX1 are ~28.03 Å and 100 Å, larger than the values of InsP₆-bound SPX (25.33 Å and 89 Å respectively). This result suggests that SPX1 is less compact than InsP₆-bound SPX1. We also calculate Chi-square for SAXS datasets +/- InsP₆ using formula as follows:

$$\chi^2(r_0, \delta\rho) = \frac{1}{N_p} \sum_{i=1}^{N_p} \left[\frac{I_e(s_i) - cI(s_i, r_0, \delta\rho)}{\sigma(s_i)} \right]^2$$

The value is 11, indicative of obvious difference of two datasets. We also included Chi-square values for our atomic models and SAXS experiment data in the legend of Supplementary Figure 10. Since R_g and D_{max} are widely used to quantitatively compare the difference of two datasets, we only put these parameters in Supplementary Figure 9.

3) Addressed – fixed very sloppy stoichiometry.

4) Addressed – fixed crystallography table and explained real space problems and rama outliers.

5) Authors still do not state syringe and sample cell buffers were MATCHED although other problems with ITC were fixed. What’s being injected via syringe must be identical to buffer in the sample cell – I’m sure they did that but they have to state that is the case. Otherwise the ITC data should not be published.

Response: Thank you for pointing this out. This has been answered in our response to A).

6) *Addressed – misleading language removed.*

7) *Addressed – mistake fixed.*

8) *Addressed – misleading language removed.*

9) *Cannot determine – The old AND new line numbers are needed to determine if new statements have been revised.*

Response: Thank you for pointing this out. This has been answered in our response to B).

10-17 were all addressed.

Reviewer #2 (Remarks to the Author):

The authors have addressed all my concerns.